# Enough to Feed Ourselves!—Food Plants in Bulgarian Rural Home Gardens

**DOI:** 10.3390/plants10112520

**Published:** 2021-11-19

**Authors:** Teodora Ivanova, Yulia Bosseva, Mihail Chervenkov, Dessislava Dimitrova

**Affiliations:** 1Department of Plant and Fungal Diversity and Resources, Institute of Biodiversity and Ecosystem Research, Bulgarian Academy of Sciences, 1113 Sofia, Bulgaria; julibos@abv.bg (Y.B.); vdmchervenkov@abv.bg (M.C.); 2Faculty of Veterinary Medicine, University of Forestry, 1797 Sofia, Bulgaria

**Keywords:** ethnobotany, subsistence, home gardens, self-provision, traditional food

## Abstract

The home garden is a unique human-nature interspace that accommodates a diverse spectrum of plant species and provides multiple services to households. One of the most important roles of home gardens is to shelter the agricultural plant diversity that provides for diverse and healthy nutrition, especially in rural communities. While tropical home gardens have received wide recognition due to their provisional function for the local communities, temperate and especially European home gardens have been discussed less frequently as a source of subsistence. The main objectives of the current study were to document plant species grown in Bulgarian rural home gardens and to explore related local knowledge and cultural practices that influence food plant diversity, its selection and preservation. Field work was focused on settlements situated in eight provinces in South and North-West Bulgaria. Participants representing 65 home gardens were approached through semi-structured interviews. Home gardens were found to harbor 145 cultivated and semi-cultivated plant taxa, used as food, medicinal and aromatic plants and as animal fodder. Members of the Rosaceae family were most numerous. The largest part of the garden area was occupied by vegetable crops of Solanaceae and Cucurbitaceae. In 63.1% of the studied households, the food growing area comprised more than 2/3 of the total size of the garden. Most preferred crops reflected the social and cultural importance of food self-provisioning, especially in the rural areas. The provisional role of the home gardens in regard to preparation of traditional foods and the driving forces for seed saving are discussed.

## 1. Introduction

Through the ages, humans have invested considerable efforts and resources to construct and maintain a favorable environment for growing and breeding crops and ornamental species so as to foster a diverse spectrum of provisions and services delivered by plants [1,2,3]. As a result, one of the closest spaces of plant–human interaction–the home garden–is seen as a multifunctional and multilayered land-use system that provides not only for production of food and fodder, ornamentals, medicines, fuel, etc., but also for human cultural and spiritual well-being [4,5]. Depending on the climate, geographical region, cultural and economic background of their keepers, home gardens differ in their structure, natural diversity and purpose [1,6,7]. Private gardens in the tropical zones have been more frequently discussed in their current relation to self-provisioning, presenting an important part of the livelihoods of economically struggling communities [8,9,10,11,12]. A broad variety of plants used as food are reported in a number of studies dedicated to the past and present of the cultivation of edible species in European home gardens [13,14,15,16,17]. However, being largely supplanted by the globalized agri-food industry, temperate and especially European home gardens and their owners have drawn less attention [5,16]. Since the beginning of the new Millennium, European home gardens have presented a colorful spectrum in East–West direction. In Western and Central Europe, they have been mostly appreciated for their cultural value, harboring mainly ornamentals and fruit trees, while to the east they have ensured staple food and/or fruits and vegetables for their owners and in some cases also for the market [7,18,19]. Important topics for researchers of European home gardens have also been the loss of agricultural diversity, preservation of biocultural heritage and the role of gardening in the distribution of invasive plant species [5,17,20,21,22,23]. However, with the bloom of urban gardening, activated by sustainability objectives and more recently by the food import shortages imposed by the Covid-19 pandemic, there is a growing body of evidence showing that (home) gardening in the industrialized societies (namely in Continental Europe, Britain and North America) is a vital activity that could not only alleviate family/local food insecurity but also contribute to the mental health and subjective wellbeing of those involved in it [8,24,25,26,27,28,29,30,31].

In Eastern and Central European countries, and especially in their rural regions, where substantial socio-economic and agricultural landscape transformations have been evident since the abolishment of Communism, the fate and development of home and allotment gardens as a food source were considerably different compared to their western counterparts [32,33]. Through the years, different studies have related subsistence farming in Eastern Europe with the overcoming of economic challenges to the poor communities in these countries [34]. However, recent data have shown that the meanings that underpin self-provisioning by growers of allotment gardens from Central and West Europe were quite similar [35,36]. Nevertheless, it has also been shown that for East Europeans, food growing is not only a simple means of food supply but also a complex culture-related activity that needs additional research [37].

Subsistence farming was a primary livelihood in Bulgaria until the collectivization of farm land after the end of World War II [38]. As a result, home gardens, and later, allotments granted to the urban population, remained the main places where Bulgarians could freely operate as farmers during Communism. The popularity of subsistence farming in countries of the former Eastern Bloc at the beginning of the 1990s was negatively correlated with the percentage of the population employed in the agriculture sector, with Bulgarians placing food growing for personal use among the main activities that supported their overall living standard [39]. A recent study has shown that in several peri-urban villages around Sofia (the capital of Bulgaria), vegetable gardens and greenhouses were found in 31–71% of the examined yards, while flower beds and lawns were less frequent (15–53%), depending on the terrain, planning and individual preference of the owners [40].

In the context of growing urbanization and a shrinking agricultural sector, it was important to understand the actual state of plant diversity grown/maintained in home gardens as well as the processes that have guided home gardening in these very dynamic socio-economic and political times accompanied by ongoing climatic changes. We focused on the food plants in home gardens to explore changes related to access to imported seeds and planting material, which has been eased since the end of Communism and the subsequent accession to the EU (2007). Our interest in food plants was additionally shaped by household food consumption analyses before and shortly after the democratization of the country in 1989, which showed very low consumption of fruits and vegetables. These habits have not changed since then [41,42]. The current paper aimed to provide evidence about food, medicinal and fodder plants grown in home gardens in rural regions in Bulgaria and to discuss the importance of ethnobotanical and horticultural knowledge that backs up the provisional as well as conservational role of these gardens.

## 2. Results

### 2.1. Characterization of Home Gardens and Their Owners

The characteristics of the studied home gardens and their owners are shown in Table 1. Home gardens sized over 100 square meters prevailed in both lowland (≤300 m above sea level, coded hereafter with LL) and semi-mountainous and mountainous settlements (>300 m a.s.l., coded hereafter with SMM), at 93.8% and 57.6%, respectively. The number of smaller gardens was higher in SMM settlements, where yards of less than 100 square meters were frequently found due to the elevated terrain (*p*-value < 0.01).

Traditionally home gardens were entirely fenced, arranged in the front and back yard, proportioned according to the position of the house and the size of the street-facing part of the yard. In the mountainous areas (e.g., Rhodope Mountains) small vegetable plots were also located outside home yards, occupying every flat space with fertile soil in the villages; these were fenced only against animals. Traditionally, orchard trees, small fruit and nut shrubs, as well as (semi)-cultivated greens such as stinging nettle (*Urtica dioica* L.), sorrel (*Rumex acetosa* L.) or garden orache (*Atriplex hortensis* L.) occupied the periphery of the gardens. In home gardens with a larger number of fruit trees, these were distributed unevenly and rarely in rows to allow cultivation of the garden’s maximum surface area. In 82.7% of the home gardens *Vitis vinifera* L. vines were trained on pergolas so to shade over the home/yard entrance, while in 13.4% of the gardens, wine grape varieties were found on trellises, in rows, similarly to local vineyards. The latter was of a specific interest to the participants producing wines and spirits for their own consumption. Only in about 10% of the gardens did we find both types of vine cultivation.

Ornamental plants typically bordered food growing plots and yard pathways, or were set in narrow strips or patches close to the house and along the fences to free the major portions of the home garden for cultivation of mainly annual edible plants, medicinal plants and spices, and occasionally fodder for animals. The food growing area (FGA) in 63.1% of the studied households comprised more than 2/3 of the total size of the garden and in most of the remaining gardens ranged between 1/3–1/2 of their total size (35.4%). The size of the home garden and FGA correlated positively in SMM settlements (Spearman *ρ* = 0.706; *p*-value < 0.01) but not in the LL ones (Spearman *ρ* = 0.273; *p*-value = 0.207). However, there was no statistically significant difference between LL and SMM groups in the FGA proportions. The larger size of the yards in LL settlements allowed garden owners not only to develop substantial FGA but also to create lawns and/or other recreational spaces such as garden arbors.

The food growing area was typically unified or separated in plots by temporary pathways. Raised beds were not found as a garden practice. Crop rotation was applied to maintain soil fertility. Cultivation plots were designed according to the irrigation arrangements on site. Irrigation of the open plots (furrow method) relied on private wells, adjacent water bodies/springs and/or tap water. To manage water shortages and high water bills, drip irrigation was maintained, yet rain-water collection systems were scarce in the visited households.

Food plant growing in facilities that provided greenhouse conditions was recorded in about half of all home gardens. The frequency of these facilities slightly prevailed in SMM (56.2%) compared to LL settlements (42.4%, *p*-value = 0.325). Cold-frames and low-tunnels were used for seed germination and growing of seedlings in late winter and early spring. However, to protect against acid rain, hail and frost, some owners preferred to cultivate tomatoes and cucumbers using plastic foil greenhouses for the whole growing season both in SMM and LL; some households had the option to cover more than 20 square meters. Regardless of the installed facilities for protected agriculture, winter was typically a resting period for gardening activities in the visited gardens. Vegetable growing in greenhouses outside the season (i.e., spring onions and lettuce in October–November) was reported by participants from southern parts of the country.

Gardens were owned by families (41.5%) or single-person households, among whom females prevailed (38.5%) compared to males (13.8%). Most of the owners were seniors (late adults) with an average age of 68.52 years, and a majority of whom were within the age range of 56–75 years (55.8%). Those in SMM settlements were slightly older. The percentage of people under 55 years of age among our participants was low (13.5%). The age and sex of the participants were not significantly associated with the size of the cultivated FGA or the size of the garden as a whole. Similarly, regardless of the variations in garden and FGA size, most of the home gardens were maintained by one (27.7%) or two (49.2%) persons. The participants shared that mechanized ploughing was used when it was possible, and all of them used agrochemicals (pesticides and fertilizers). In this sense, the home gardens were more or less arable land that provided all plant produce for the household except for wheat grain. Our participants also often referred to as “gardens” any (mainly irrigated) plots outside the settlements that they used to cultivate through the years to supply additional food. However, no one regarded themselves as a “gardener”, as this term remains reserved for occupational gardeners. Almost all participants shared that they had some involvement in industrial agriculture currently or in the past, due to the obligatory participation of students and workers in agricultural activities required by the state before 1989; however, only two participants had specialized training in agriculture or horticulture.

### 2.2. Plant Diversity and Crop Preference

We documented in Bulgarian rural home gardens 145 cultivated and semi-cultivated plant taxa used as food, medicinal and aromatic plants, and as animal fodder. A list of all recorded taxa is presented in Appendix A. They belonged to 40 plant families and 98 genera, with members of the Rosaceae family being the most numerous (22 species, 15.2%). The following plant families were less represented: Lamiaceae with 14 taxa (9.7%), Fabaceae (12 taxa, 8,3%), Amaryllidaceae (11 taxa, 7.7%) and Asteraceae (10, 6.9%). *Prunus* and *Allium* were the richest in taxonomic genera cultivated in the home gardens (9 taxa each), followed by *Cucurbita*, *Ribes*, *Beta*, and *Brassica* with 4 taxa each.

Fruits were the most frequently used plant parts (35.9%), followed by leaves (24.1%). Whole herbage (17.6%), mainly of culinary herbs, leafy vegetables and/or medicinal plants, was used for salads, herbal teas and/or as pastry fillings. For the preparation of pastries, ruderal species such as *Papaver rhoeas* L., *Capsella bursa-pastoris* (L.) Medik., *Stellaria media* (L.) Vill., and *Ranunculus arvensis* L. were carefully preserved while juvenile in the garden wherever they appeared, whereas other semi-cultivated species such as *Portulaca oleracea* L., *Rumex patientia* L., *Urtica dioica* L., and *Phytolacca americana* L. were kept where they did not interfere much, e.g., along the fences, among the vegetables, or in the flower bed. Seeds of 14.1% of the taxa were used as staple foods (legumes), snacks (legumes, cucurbits, sunflower, hazelnuts and peanuts), or as spices (the representatives of Apiaceae—dill, fennel, parsley, lovage, and coriander).

There was no significant difference in the average total number of taxa cultivated in the home gardens for food, medicinal purposes or fodder when LL and SMM settlements were compared (Table 2, Fisher’s exact tests *p*-values > 0.05). Home gardens in LL settlements tended to comprise a higher number of crop taxa. However, the garden richest in taxa (53) was in an SMM region, had been recently enlarged by merging with a neighboring yard, and was also close to the Bulgarian-Greek border, hence under Mediterranean climate influence. In all gardens, at least 10 cultivated taxa were edible, but in most cases this number was over 20. Plants used as culinary herbs were often used also as medicines. Nineteen plant species grown for medicinal purposes were employed as herbal teas: e.g., *Agastache foeniculum* (Pursh) Kuntze, *Sideritis scardica* Gris., *Melissa officinalis* L., *Ocimum basilicum* L., *Tilia tomentosa* Moench and another six for extracts (e.g., *Valeriana officinalis* L., *Lilium candidum* L., *Juglans regia* L., *Allium ursinum* L.). They were applied as remedies for cough, stomach and intestinal discomfort, relief of mild nervous tension, skin conditions, etc.

A small number of the cultivated taxa (19) occurred in all the provinces, and only six were found in every visited garden (Table 3). The fruiting and bulbous (mainly onion and garlic) vegetables and culinary herbs were the groups that were obligatory for all home gardens. The first two groups occupied the greater part of the gardens as well as most of the greenhouses. While winter white and yellow onion varieties (*A. cepa* L. Cepa Group) and garlic (*A. sativum* L.) were regarded as “ordinary”, red onion varieties (*A. cepa* L. Cepa Group) were of explicit interest in several regions where local landraces of this crop were grown under the name of the specific village, e.g., ‘Peshternski luk’ (Peshterna village, Lovech Province), ‘Reseleshki luk’ (Reselets village, Pleven Province) and ‘Banichanski kromid’ (Banichan village, Blagoevgrad Province). The specific and appreciated quality and taste of these landraces were attributed to the local water sources. Red onion varieties were used for preparation of traditional foods typical for the village or the region and were celebrated on occasions with specially organized local festivities related to their harvest.

The remaining *Allium* species (*Allium tuberosum* Rottler ex Spreng., *A. schoenoprasum* L. and *A. vineale* L.) were far less typical for the home gardens, with the latter two used only in salads. Consumption of fresh salads in the winter–early spring period was also a main driver for cultivation of lettuce (*Lactuca sativa* var. *longifolia* (Lam.) Alef. and *Lactuca sativa* var. *crispa* (L.) Schübl. & Martens) and radishes (*Raphanus raphanistrum* subsp. *sativus* (L.) Domin). *Brassica* varieties were used fresh and also as major ingredients for pickles.

“What we grow in the garden? You know, those who never eat peppers and tomatoes will never grow up”, replied N.N. (70, male), paraphrasing the beginning of a popular Bulgarian children’s song about healthy eating habits and the importance of eating vegetables. Indeed, fruiting vegetables, and specifically those used in popular salads and vegetable dishes, were of primary interest to our participants. Most of their investments in greenhouse construction, irrigation systems and seedling maintenance were dedicated to the cultivation of crops from Solanaceae and Cucurbitaceae. In the LL settlements, fruiting vegetables frequently occupied 1/5–1/3 of the FGA, comparatively more than in SMM gardens, or 81.3% and 57.6% of the gardens, respectively (Figure 1a,b; Appendix A; Fisher’s exact tests *p*-values < 0.05).

Cultivars of *Capsicum annuum* L. were found to be most diversely consumed—raw or roasted in salads, in cooked dishes, or as a snack; stored frozen or in preserves such as the traditional relish *lyutenitsa* (together with tomatoes, aubergines, carrots, etc.) or in pickles; as well as for preparation of sweet paprika and chili powder.

There was no significant difference between LL and SMM settlements in preference for cultivation of the remaining food plant groups. Herbs and spices (29 taxa), arranged near ornamental patches or in between vegetable plots, mostly occupied less than 1/5 of the FGA both in LL and SMM settlements (Fisher’s exact tests *p*-values > 0.05). Similarly, 60–90% of all participants cultivated legumes, tuber and root vegetables, leaf and stem vegetables, and berries and small fruits on less than 1/5 of the FGA (Fisher’s exact tests *p*-values > 0.05). Still, home gardens with poor soils and irrigation opportunities (SMM settlements) were more likely to harbor slightly larger plots with legumes, tuber and root vegetables.

Cultivated plots had very dynamic use, and their owners had detailed plans for the first and second croppings throughout the FGA. Leafy greens, together with spring onions, usually occupied the greenhouses and sheltered plots before the start of the active growing season (February–April). Consequently, they were then replaced by fruiting vegetables, potatoes or legumes. To maintain maximum utilization of the FGA, all unsuccessful plants were immediately substituted with seeds or seedlings bought from the nearest farmers’ market or obtained from friends and relatives. This was mentioned as specifically important for garden owners using tap water for irrigation.

Legumes were represented by eight species. Apart from the common bean, *Phaseolus vulgaris* L., which was most popular in the home gardens, five pre-Columbian staples, i.e., fava beans (*Vicia faba* L.), grass pea (*Lathyrus sativus* L.), chickpea (*Cicer arietinum* L.), lentil (*Lens culinaris* Medik.) and cowpea (*Vigna unguiculata* (L.) Walp.), were scarcely grown in gardens in southeastern parts of the country. Those were maintained only for the preparation of a few traditional foods, hence their limited use had reduced them to garden crops. Conversely, interest in pole bean varieties and landraces (*Ph. vulgaris*, *Phaseolus coccineus* L. and in some areas, *Phaseolus lunatus* L.) was higher as they occupied less space and allowed intercropping with corn and Cucurbits. Recently, these varieties have attracted higher consumer interest due to their large seeds, which are preferred in various salads.

Corn (*Zea mays* L.) was the only species representing Poaceae in the home gardens. In more than half of LL settlements, corn was missing, and in the rest, it occupied less than 1/5 of the FGA. Corn was more frequently sown in SMM settlements where local landraces, specifically used for preparation of cornbread and typical dishes, were found in very limited areas for preparation of festive foods for community occasions and for preparation of specialties in local restaurants. Our participants from the Rhodope Mountains shared that reduced growing of corn, along with other grains such as rye (*Secale cereale* L.), was related to the improved access to wheat flours and corn produced in other regions.

Fodder plants were the least represented group in the home gardens. They were preferred by owners breeding farm animals within their home premises to ensure easy access and to reduce transportation of the fodder. *Medicago sativa* L. was the only one sown specifically as fodder. Corn, fodder beet and herbage of various legumes were also used for the same purpose.

Apples (*Malus domestica* Borkh.) and sweet cherries (*Prunus avium* (L.) L.) were found to be the most favored fruit trees across the country. They were followed by pears (*Pyrus communis* L.) and *Prunus* taxa. Pomoideae and Prunoideae were typically found to range from 3 to 10 individuals per garden, and only keen grafting enthusiasts maintained more varieties and individuals in their home gardens (Table 4). Several species introduced from the Mediterranean region and subtropical Asia were found to be popular all over the country, i.e., fig (*Ficus carica* L., 27 gardens), kiwifruit (*Actinidia deliciosa* (A.Chev.) C.F.Liang & A.R.Ferguson, 11) and lemon trees (*Citrus limon* (L.) Osbeck, 17). The latter were predominantly sheltered for the winter period. In the southern parts of the country, with Mediterranean climatic influence, there was a greater variety of such species: *Ziziphus jujuba* Mill. (15), *Laurus nobilis* L. (9), *Eriobotrya japonica* (Thunb.) Lindl., (7), *Prunus dulcis* (Mill.) D.A.Webb (7), *Punica granatum* L. (6), *Diospyros kaki* L.f. (5), and *Morus nigra* L. (1). *Juglans regia*, *Castanea sativa* Mill. and *Corylus* species were found less frequently due to their incompatibility with the vegetable plots.

### 2.3. Seed Saving and Plant Food Self-Provisioning

Food provisioning requires access to seeds and other planting materials, as well as relevant skills in horticulture. Thus, the abilities to collect quality seeds, to produce healthy seedlings and to obtain larger yields were considered central for the creation of a “successful” rural life. Home-grown crops and homemade foods were claimed to be more valuable and of higher quality compared to purchased ones. Seed saving of crop varieties was practiced by 60% of the participants, but they had various attitudes regarding seed acquisition and conservation. In terms of origin, three sources of seeds were recorded–marketed seeds produced in Bulgaria, imported seeds and heirloom seeds/local landraces obtained outside the market. Interestingly seed saving was not significantly related to food self-provisioning, and those who were interested in seed saving were targeting all kinds of seeds regardless of their origin (*p*-value < 0.05, Table 5).

As the main focus was placed on sufficient and quality harvests, the origin of the crop varieties was not imperative. Seeds obtained from the national market were claimed as least favored for ensuring quality seeds for saving (*p*-value < 0.05). Very often, garden owners were disappointed by the quality of the seeds obtained from Bulgarian markets (low or uneven germination, variety or even species mislabeling). Seeds found available on the Bulgarian markets were mainly of hybrid (F1) origin, due to which seed saving was deemed unsuccessful in most cases. If the saved seeds gave satisfactory harvests in the second year, they were kept, reproduced and gradually perceived as “owned” seeds. Very often, garden owners who were respected for their skills by their communities had specific interest in saving seeds from local landraces, which they also spread among their fellow villagers.

Nowadays, there is growing consumer interest in “traditional” tastes; hence, the cultivation of heirloom seeds has become popular among garden owners. However, only two of our participants kept records of the origin of their crops, focusing rather on the skills needed to grow food plants than on the preservation of local landraces and/or heirloom varieties. The cultural and ecological values of local landraces were not entirely understood, especially by seniors that regarded local plant diversity as granted. Some garden owners had higher trust in imported seeds, especially when the taste of the obtained produce was close to their expectation.

Adding variety to the cultivated crops in the home garden was rarely considered as posing a risk of introducing invasive species or eroding local plant genetic resources. The participants had no opinion or were hardly interested in plant conservation as a grassroots activity/responsibility. In some cases, local cultural centers took the initiative to investigate and collect the knowledge about local landraces and occasionally created small seed banks representative for the village. A push forward for such activities was the collaboration of the cultural centers with Slow Food, an international organization maintaining a worldwide online catalogue (Ark of Taste) of threatened foods, among which a considerable share was occupied by fruit and vegetable varieties and preserves from them. The pink tomato from Kurtovo Konare, Kurtovka apple, Smilyan beans and others are part of the Ark.

During the interviews, seed saving was described as mainly female activity, performed by elderly women, grandmothers and/or mothers-in-law. The informal exchange of seeds was favored especially when the provider or his/her ancestors were known as skillful garden keepers.

Seeds from crops of Solanaceae and Cucurbitaceae were collected from specifically selected fruits, while those from legumes and corn were collected from healthy, strong plants to avoid transmission of diseases for the next year. Seeds were stored in various repurposed containers, envelopes and jars. Home “seed banks” were kept in storage rooms in cool and dark places. *Mentha spicata* L., *Origanum vulgare* subsp. *hirtum* (Link) Ietsw. and *Juglans regia* leaves and salt were used as traditional insect repellents.

Most of the studied households (86%) relied entirely on their home gardens for the provision of plant produce (except wheat grain) for the whole year (Table 6). Most participants from the lowlands (93.3%), who also managed bigger gardens, produced enough to be independent from the market, while some of those from SMM areas (22%) still relied on the market due to the smaller garden size. Larger home gardens and FGAs, logically, allowed their owners to process higher quantities of their produce into fruit and vegetable preserves for the winter or to offer the surplus for exchange, tastings at local events, or on the local farmers’ market (Chi-S *p*-values = 0.029 and 0.036, respectively).

Few participants who used their properties only as summer villas opted only for fresh produce for the summer (13.8%). Our participants invested substantial time and effort into processing the produce from their gardens by drying, canning and pickling in order to store it for the winter. Preparation of canned food included not only preserved vegetables and fruits but also ready meals and starters that were not found on the market or were not of the desired quality and/or local taste. Preserved plant foods were consumed seasonally, in winter–early spring, and consequently replaced by the fresh early leafy vegetables (lettuce, spinach, radishes, etc.). Especially older participants, living alone, preferred to utilize most of their harvest by processing, thus providing food not only for themselves but also for their children, other relatives and friends.

Food self-provisioning from the garden was perceived by the participants as an almost inevitable activity in a rural community. However, it ensured not only self-sufficiency but also catered to their personal pride and recognition within their community. Quality non-perishable foods were shared within the extended family and the whole community and also served as customary gifts for guests and for traditional holidays. Thus, food produced in home gardens not only gained added value as gratification for the people’s effort but also performed important social and cultural functions. Occasionally, the produce surpluses were offered for sale to visitors of the villages, at local markets and, when organized, at local fests. Six of our participants regularly offered fresh produce (fruits, onion, and leafy greens) at the local farmers’ markets or to wholesalers who visited their villages. These endeavors also involved only surpluses. Garden produce was also exchanged among fellow villagers to ensure more diverse plant foods. In some cases, home gardens (5) served for the provisioning of guest houses by their owners, thus adding value to the tourist service and ensuring the authenticity of the “taste of place”. Food from home gardens was considered of high quality by their owners mostly because it corresponded to their expectations for tastes that have deep roots in their memories and traditions. It also strengthens the ties between generations and serves as a vector for transmission of gastronomic knowledge among them.

The importance of some of the widely distributed and well established garden food plants was celebrated at local festivals, an activity that has recently become well-known. Among many, particularly popular are the Kurtovo Konare Fest dedicated to local plant food diversity and horticultural traditions and the Festival of Smilyan Beans that celebrates a complex of bean varieties cultivated in the Upper Arda Valley in the Rhodope Mountains. These festivals demonstrate the local food diversity and gastronomic skills and have attracted a growing number of visitors for over a decade. They have become a central part in the development of Kurtovo Konare and Smilyan as tourist destinations.

## 3. Discussion

The current paper provides evidence for the crop diversity of home gardens in LL and SMM regions of Bulgaria, the significance of home garden for the self-provisioning of food in rural areas, and their function as a space for biocultural interactions and transmission of local traditional knowledge. The most specific characteristics of the studied home gardens were the presence of definitive boundaries and the predominance of the area used for growing food crops for self-provisioning, i.e., they belong to the category of subsistence gardens [4]. Our participants preferred to maintain a larger portion of their gardens undivided, thus creating an arable field-like system that provides for a variety of plant foods for the household except for wheat grain. This allows higher plasticity in the selection and cultivation of crops in comparison to garden cultivation practices currently popular in Western Europe (e.g., raised beds, pots, tubs) [25]. Ethnobotanical analysis of the Bulgarian folk songs (collected in the late 19th and early 20th centuries) presented Bulgarian home gardens as herbal or ornamental plots around the house where a limited number of plant species were cultivated [43]. Sociological and landscape studies have shown that, before land collectivization in 1945, home gardens in mountainous areas provided limited plant produce as their owners were mostly livestock breeders and artisans, while in the lowlands, growing food plants was more typical for garden plots in the outskirts of the settlements [44,45]. The transformation of the garden during the last century could be related to the traditional market-oriented gardening practiced by Bulgarians abroad until the 1940s and to the utilization of advancements in agronomy picked up from the state agricultural complexes that managed agriculture in Bulgaria during Communism. These skills and practices were down-scaled for subsistence purposes in the home gardens [34,46]. Studied gardens also partially resembled the allotment gardening practiced in the periphery of urban spaces across Europe [35,47]. However, in Western Europe (Germany) only about half of the participants in urban gardening projects were motivated by self-provisioning of food [25]. The same authors showed that topics such as organic farming practices driven by the quest to reconnect with nature, a variety of composting techniques, soil protection and improvement, mulching and crop rotations, and renewable energy technologies were also of interest to urban gardeners.

The cultivation practices applied by our respondents were labor- and resource-intensive, thus presenting Bulgarian rural home gardens as relatively large, high-maintenance spaces, contrary to the approaches favored by garden owners in the Austrian Alps, who sought to reduce additional labor input when expanding their gardens [16]. Those and also many home gardens around the world were reported to depend less on inputs in mechanization and industrial agrochemical products, thus enhancing their role as sustainable sources of healthy food [8]. Considering the age of our respondents and the predominately female participation in our sample, we could conclude that women were decision-makers and the main workers in Bulgarian home gardens, which could explain the large number of cultivated crops, similarly to previous reports [18,22]. The number of cultivated and semi-cultivated taxa in Bulgarian rural home gardens that were used for food and as medicinal and/or culinary herbs was rather high. However, the number of species per garden ranged similarly to previously reported data for home gardens in other European countries [16,18,48]. Rosaceae was the most diversely represented family, but the fruiting vegetables from Solanaceae and Cucurbitaceae were of main interest to our participants. The most-preferred food plants formed a set of six taxa present in every garden (*Allium cepa*, *A. sativum*, *Capsicum annuum*, *Cucumis sativus* L., *Lycopersicon esculentum* Mill., *Petroselinum crispum* (Mill.) Fuss). Such crop preference was typical for all gardens regardless of the size of the garden and FGA, settlement elevation, or soil and climatic differences, which suggests culture-specific selection of garden crops. The elevation of the home garden was shown to influence plant/crop diversity in Ugandan home gardens, but such comparisons were not targeted in studies of home gardens in Europe [49]. However, *Capsicum annuum* and *Lycopersicon esculentum* were most popular also in mountainous areas of the Iberian Peninsula [18].

The provisional functionality of Bulgarian rural home gardens corresponded with those of home gardens in Hungary, Spain and Turkey, where plant produce cultivation was also a primary incentive [18,50,51]. The currently studied home gardens were found to be almost entirely dedicated to the production of staples and vegetables for human consumption with five or less plant species used as culinary herbs and/or medicines. Fodder cultivation was found in a very low number of gardens, even in SMM settlements, corresponding with a reduction in the number of animal farms in Bulgaria in recent years [52]. In Europe (Eastern Alps and Hungary), the multifunctional profile of the garden was related to the production of fodder for livestock [16,50]. Tropical home gardens also were found to provide not only for more diverse and healthy nutrition but also for firewood, timber and cash crops [9,53]. The crops preferentially grown in Bulgarian home gardens were used in various traditional foods shared within the family and the community. They reflected the social and cultural importance of food self-provisioning, especially in the rural areas. Focusing on food crops for self-provisioning on the farm/in the home garden was regarded for many years as an approach to alleviate economic insecurities in Central and (South) Eastern Europe. However, recent reports have shown that this practice has multiple examples throughout Europe, both in subsistence and non-subsistence establishments [37,54,55,56,57,58,59,60,61]. Our respondents associated food self-provisioning from their gardens also with access to quality food and with overall positive experiences related to the exchange of (plant) resources and skills in horticulture.

Given the broad range of cultivated edible taxa we documented in rural home gardens, one could argue that the food provisioning focus of home gardens reduces plant diversity, leading to specialization as seen in other regions [62,63]. In this sense, local cuisines should be mentioned as an important driving force for preservation of traditional ecological knowledge about wild edible plants and local landraces [64,65,66,67,68,69]. The obligatory crop mixes of Bulgarian rural home gardens, together with the remaining and most popular taxa, were well-aligned with the preparation of most of the traditional foods in Bulgaria and provided very well for the recommended plant foods typical of the Mediterranean diet [70]. Strictly speaking, the home gardens we assessed could not provide for the breads that are viewed as the most important plant foods typical of Bulgarian cuisine, including ritual food [71]. Here it is important to mention, however, that none of the eight species of legumes we found in the studied home gardens was part of the obligatory mix, and only *Ph. vulgaris* was present in all provinces. The role of the pre-Columbian pulses was limited to the southernmost parts of the country, a region with typically hot summers and frequent droughts [72]. Given the well-known drought susceptibility of *Ph. vulgaris* [73,74], our data imply that these pre-Columbian legumes were preserved in the home gardens as drought-resistant pulses. Interestingly, growing the same species was also cited as risk management practiced in these territories for millennia, as evident from archaeobotanical studies of Bulgarian Early Neolithic sites [15].

Preserving edible ruderals and local landraces in the home gardens also could be regarded as evidence, although with fading significance in the context of growing urbanization, for the importance of home gardening in supplying specific ingredients for the preparation of traditional foods that are otherwise impossible or hard to find on the market [50,75]. The variety of plant parts used and their methods of preparation also testify to the importance of garden food plants in the lives of local people and demonstrate the role of the garden as a crossroads of memories, knowledge and modern inventions for their owners [50,76].

Trading in local plant genetic diversity for securing better yields should be mentioned as a negative trend in Bulgarian rural home gardens. Abandoning specific characteristics of local foods not only deprives them of their identity [77] but also disrupts two of the most important functions of home gardens: to serve as an initial ground for plant domestication and as a space for preservation of plant genetic resources [5,75,78]. While in recent years rural communities in some developing countries received due attention and support in this direction and many European communities seek to cooperate with their efforts in preservation of local agrobiodiversity [79,80], their Bulgarian counterparts should act urgently to benefit from the local crop diversity that has remained fairly preserved in the rural regions. The latter also stresses the importance of broader awareness campaigns and horticulture literacy promotion measures needed in the industrialized countries, where the transmission of traditional knowledge has weakened rapidly due to depopulation of rural areas and growing urbanization.

## 4. Materials and Methods

### 4.1. Study Area

Bulgaria is located in Southeast Europe, covering an area of nearly 111,000 square kilometers. The territory elevation ranges from sea level to nearly 3000 m, with the Balkan Mountain range spanning transversely from west to east, dividing the country into northern and southern parts. Bulgaria is in the transitional region between temperate and Mediterranean climatic zones. Average temperatures vary between −7° and 3 °C in the winter and 10°–25 °C in the summer, with climate becoming warmer and dryer since the end of the twentieth century [72]. Average annual values for precipitation range between 450–1100 mm, with drought periods at the end of the summer [72]. Bulgarian vascular flora are diverse and comprise 4064 species of spermatophytes affiliated with 921 genera and 159 families [81]. The arable land comprises about 50% of the country’s territory, while forests account for 42% [82]. We selected study targets in both the southern and northern parts of Bulgaria in order to cover the diverse topological and climatic characteristics of the country.

### 4.2. Field Study and Research Approach

The field study (2017–2021) was carried out with the participation of the heads of 65 households in eight Bulgarian provinces (Blagoevgrad, Haskovo, Plovdiv, and Smolyan in the south and Lovech, Montana, Pleven and Vratsa in the north). The selection of participants (Table 1) was random or assisted by informal community leaders. Gardens, classified into two groups according to the elevation of the settlements—lowland settlements (LL) from sea level to 300 m a.s.l. and semi-mountainous and mountainous settlements (SSM) above 300 m a.s.l., were visited at least once in the growing season. Semi-structured interviews were performed in order to collect local knowledge about cultivated plants in rural home gardens used for food, spices and medicines and fodder for backyard animals. Participants were asked to share information about the local food and medicinal plant names, crop preferences, established cultivation and seed-saving practices and the exchange of planting materials in the past and present. Personal impressions/perceptions on the changes in their gardening endeavors through the years were also recorded. Informed consent was verbally obtained from every participant prior to the interviews. The guidelines prescribed in the Code of Ethics of the International Society of Ethnobiology [83] were followed during the field study, and their compliance was confirmed by the Scientific Council of the Institute of Biodiversity and Ecosystem Research, Bulgarian Academy of Sciences, acting as independent institutional Ethics Board (Decision No. 6/21/05/21).

The following characteristics of home gardens were assessed: size in square meters, presence of greenhouse facilities for growing food plants, size of open food growing area (FGA), expressed as a fraction of the home garden size, and size of the plots occupied by different crop groups as fractions of the FGA. The latter was adopted due to the use of large areas of the gardens for different early- and late-season crops in one and the same year. Cereals, fruiting vegetables, bulbous plants, leafy vegetables, culinary herbs and spices, medicinal plants and fodder were accounted for as fractions of FGA. Trees, shrubs and vines were counted individually.

Participants were described by their age, sex of the gardener(s) and number of the people engaged in the garden’s cultivation. Food provisioning from the home garden was assessed by the availability of fresh produce and preparation of canned/dried foods prepared for the winter period (0—none, 1—seasonal/only fresh fruits and vegetables, 2—fresh fruits and vegetables and plant food processing). Three categories (0—no, 1—occasional and 2—regular) were used for the evaluation of the participants’ preferences related to sourcing of seeds: seed saving, preference for the use of seeds marketed in Bulgaria and preference for imported seeds. Involvement in saving local landrace/heirloom seeds was classified as follows: 0—none, 1—some crops and 2—most crops grown in the home garden.

Image data and/or reference specimens were collected for identification purposes; herbarium specimens were deposited in the Herbarium of the Institute of Biodiversity and Ecosystem Research, Bulgarian Academy of Sciences (SOM). Identification of the plants was carried out at least to the genus and species taxonomical levels in accordance with [84]. Subspecies or varieties were determined when possible. *Allium cepa* varieties and local landraces were placed in informal horticultural groups (here, namely Cepa, Aggregatum and Proliferum) to improve data usability as suggested in [85]. Official nomenclature of the International Code for the Nomenclature of Cultivated Plants (ICNCP), as acknowledged by the International Code of Nomenclature for algae, fungi and plants (ICBN), was followed.

### 4.3. Statistical Analyses

Studied variables were found to significantly deviate from a normal distribution (graphical examination by Q-Q plots); hence, the results of comparative non-parametric tests were reported. Absolute frequencies and percentages were used to represent ordinal or nominal variables (sex, age range, home garden size, FGA and number of people engaged in garden maintenance); scale variables (age) were presented as the median. The mean values and standard deviation (SD) as well as median, minimum and maximum values were used to present the number of plant taxa grown in home gardens.

Statistical association between nominal and ordinal variables was evaluated through Chi-Square tests (Chi-S), and partial correlation (Spearman rank-order correlation coefficient (*ρ*)) adjusted for the corresponding confounding variables of settlement elevation, FGA and garden size was applied to explore participants’ preferences in seed saving and self-provisioning. Kruskal–Wallis (K-W) test was applied for comparisons across age ranges, sex and number of participants involved in the maintenance of the garden.

All statistical tests were based on two-sided test significance and with a significant level α = 0.05. All analyses were performed using SPSS (ver. 20; IBM Corp., New York, NY, USA).

## Figures and Tables

**Figure 1 plants-10-02520-f001:**
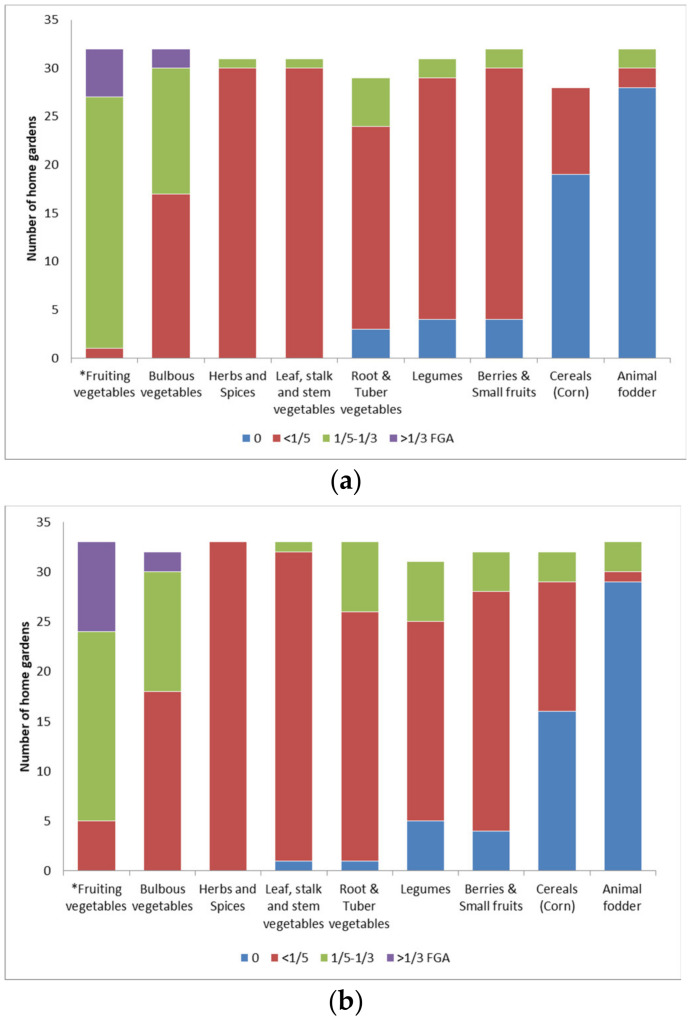
Preference given to crop groups as parts of the food growing area (FGA) across studied home gardens in: (**a**) lowland settlements (LL), *n* = 31; (**b**) semi-mountainous and mountainous settlements (SMM), *n* = 33. * Fisher’s exact tests *p*-values ≤ 0.05.

**Table 1 plants-10-02520-t001:** Characteristics of the studied home gardens and owners.

	Frequencies ^1^			*p*-Values ^2^
	Total	LL	SMM	
Total N of home gardens	65	32	33	
Home garden size (square meters)				
Over 500; N (%)	27 (41.5)	18 (56.3)	9 (27.3)	<0.01
100–500; N (%)	22 (33.8)	12 (37.5)	10 (30.3)	NS
50–100; N (%)	10 (15.4)	2 (6.3)	8 (24.2)	<0.01
Under 50 (sq. m) N (%)	6 (9.2)	0 (0.0)	6 (9.2)	<0.01
Food growing area (FGA); fraction of garden size				
Under 1/3; N (%)	1 (1.5)	0 (0.0)	1 (3.0)	NS
1/3–1/2; N (%)	23 (35.4)	12 (37.5)	11 (33.3)	NS
Over 2/3; N (%)	41 (63.1)	20 (62.5)	21 (63.6)	NS
Greenhouse				
Yes; N (%)	33 (50.7)	14 (42.4)	19 (57.5)	NS
No; N (%)	32 (49.2)	18 (56.2)	14 (43.7)	NS
Owner sex				NS ^3^
Single female; N (%)	25 (38.5)	12 (37.5)	13 (39.4)	
Single male; N (%)	9 (13.8)	2 (6.3)	7 (21.2)	
Family; N (%)	27 (41.5)	16 (50.0)	11 (33.3)	
(N ^4^)	61	30	31	
Owner age (years)				NS
Median	70.00	68.00	72.00	
Mean ± SD	68.52 ± 10.99	65.96 ± 12.37	71.28 ± 8.70	
Age range (years)	35–84	35–84	52–84	NS
35–55 N (%)	7 (13.5)	5 (15.6)	2 (6.0)	
56–75 N (%)	29 (55.8)	15 (46.8)	14 (42.3)	
76–84 N (%)	16 (24.6)	7 (21.9)	9 (27.2)	
Persons involved in gardening per home garden (number)				NS
1; N (%)	18 (27.7)	8 (25.0)	10 (30.3)	
2; N (%)	23 (49.2)	17 (53.1)	15 (45.5)	
3 or more; N (%)	15 (23)	7 (21.9)	7 (21.1)	

^1^ Home gardens are grouped according to the settlement elevation: LL, lowland settlements ≤300 m a.s.l.; SMM, semi-mountainous or mountainous settlements over 300 m a.s.l. ^2^ Fisher’s exact tests were used to compare the characteristics of home gardens of LL and SMM groups. ^3^ Kruskal–Wallis (K-W) test was applied to assess owners’ characteristics (sex, age and persons involved in gardening per home garden) in relation to garden size and FGA. ^4^ Sample sizes vary due to missing data in the different variables. NS: not significant (*p*-value > 0.05). *N*: Sample size; sq. m: square meters.

**Table 2 plants-10-02520-t002:** Plant taxa used for food, medicinal plants or fodder cultivated in rural home gardens in Bulgaria.

	Total	LL ^1^	SMM ^1^	*p*-Value ^2^
Food plants				
Mean number ± SD	22.22 ± 7.15	22.42 ± 6.59	22.00 ± 7.80	NS
Median	21.50	21.00	22.00	
Min	10.00	10.00	12.00	
Max	45.00	36.00	45.00	
Culinary herbs				
Mean number ± SD	5.16 ± 2.59	5.36 ± 3.05	4.94 ± 2.02	NS
Median	5.00	5.00	5.00	
Min	2.00	2.00	2.00	
Max	9.00	15.00	9.00	
Medicinal plants				
Mean number ± SD	2.53 ± 1.87	2.91 ± 1.87	2.17 ± 1.83	0.01
Median	2.00	2.50	1.00	
Min	1.00	1.00	1.00	
Max	6.00	7.00	6.00	
Medicinal plants (no culinary use)				
Mean number ± SD	1.16 ± 1.35	1.33 ± 1.38	0.97 ± 1.30	NS
Median	1.00	1.00	0.00	
Min	0.00	0.00	0.00	
Max	4.00	4.00	4.00	
Animal Fodder				
Mean number ± SD	1 ± 0	1 ± 0	1 ± 0	NS
Median	1	1	1	
Min	0	0	0	
Max	1	1	1	
Total				
Mean number ± SD	27.33 ± 8.78	27.79 ± 8.64	26.84 ± 9.03	NS
Median	26.50	28.00	26.00	
Min	14.00	14.00	14.00	
Max	53.00	45.00	53.00	

^1^ Home gardens are grouped according to the settlement elevation: LL, lowland settlement ≤300 m a.s.l.; SMM, semi-mountainous or mountainous settlement over 300 m a.s.l. ^2^ Fisher’s exact tests were used to compare data on home gardens of LL and SMM groups. NS: not significant (*p*-value over 0.05). SD: standard deviation.

**Table 3 plants-10-02520-t003:** Plant taxa cultivated in rural home gardens in all studied provinces.

Plant Taxon	Consumption	Preservation ^1^	Occurrence, % Gardens ^2^
*Allium cepa* L. Cepa Group	raw or cooked, pastries, preserves	P	100
*Allium sativum* L.	raw or cooked, preserves	P	100
*Capsicum annuum* L.	raw, cooked or roasted, pickles, paprika powder	D, P, F, FR	100
*Cucumis sativus* L.	fresh, preserves, pickles	P, F	100
*Lycopersicon esculentum* Mill.	raw or cooked, preserves, pickles	D, P, F	100
*Petroselinum crispum* (Mill.) Fuss	flavoring for various dishes and pickles	D, FR	100
*Cucurbita pepo* L.	raw and cooked, preserves, spirits	P	98.5
*Raphanus raphanistrum* subsp. *sativus* (L.) Domin	raw (salads)		98.5
*Solanum tuberosum* L.	cooked or roasted		80.0
*Vitis vinifera* L.	fresh, preserves, raisins, vinegar, spirits	D, P, F	80.0
*Malus domestica* Borkh.	raw or cooked, preserves, vinegar, pastries, spirits	D, P, F	73.8
*Mentha spicata* L.	flavoring for various dishes, pastries	D, FR	72.3
*Lactuca sativa* var. *longifolia* (Lam.) Alef.	raw (salads), pastries		69.2
*Phaseolus vulgaris* L.	cooked	D	67.7
*Lactuca sativa* var. *crispa* (L.) Schübl. & G. Martens	raw (salads)		61.5
*Prunus avium* (L.) L.	fresh, desserts, preserves, spirits	P	47.7
*Rubus idaeus* L.	fresh, desserts, preserves	P, FR	47.7
*Satureja hortensis* L.	flavoring for various dishes and cured sausages	D	43.1
*Ficus carica* L.	fresh, preserves, spirits	D, P	41.5

^1^ Preservation modes: D, drying; P, preserves (sterilized, marinated or boiled); F, fermented; FR, frozen. ^2^ Number of home gardens (*N* = 65).

**Table 4 plants-10-02520-t004:** Occurrence of trees, shrubs and vines cultivated in rural home gardens.

	LL ^1^	SMM
**Pome and Stone fruits** **(Pomoideae & Prunoideae)**		
0; N (%)	2 (6.3)	1 (3.0)
1–2 individuals; N (%)	1 (3.1)	3 (9.1)
3–10 individuals; N (%)	21 (65.6)	23 (69.7)
Over 10 individuals; N (%)	7 (21.9)	5 (15.2)
**Mediterranean and Subtropical taxa**		
0; N (%)	7 (21.9)	16 (48.5)
1–2 individuals; N (%)	16 (50.0)	8 (24.2)
3–10 individuals; N (%)	6 (18.8)	9 (27.3)
Over 10 individuals N (%)	2 (6.3)	16 (48.5)
**Nuts (*Juglans regia* L.,** ***Castanea sativa* Mill., *Corylus* sp.)**		
0; N (%)	23 (71.9)	14 (42.4)
1–2 individuals; N (%)	6 (18.8)	13 (39.4)
3–10 individuals; N (%)	1 (3.1)	5 (15.2)
Over 10 individuals N (%)	1 (3.1)	0 (0.0)

^1^ Home gardens are grouped according to the settlement elevation; *N*: number of gardens.

**Table 5 plants-10-02520-t005:** Personal preferences for seed saving in relation to food self-provisioning.

	Seed Saving	Bulgarian Marketed Seeds	Imported Seeds	Local Landraces/ Heirloom Seeds Outside the Market
Food-provisioning ^1^	−0.065	−0.064	−0.096	−0.007
*p*-value	0.633	0.636	0.480	0.959
Seed saving		−0.265	0.412	0.518
*p*-value		**0.047**	**0.001**	**<0.001**
Bulgarian marketed seeds			−0.293	−0.435
*p*-value			**0.027**	**0.001**
Imported seeds				0.335
*p*-value				**0.011**

^1^ Partial correlation (Spearman rank-order correlation coefficients (*ρ*); Significance *p*-value ≤ 0.05 (bold).

**Table 6 plants-10-02520-t006:** Households self-provisioning plant produce entirely by home gardening (except wheat grain), %.

	Total Average ± SD	LL ^1^	SMM	Fisher Exact Test (*p*-Value ^3^)
Total	86.18 ± 15.20	93.33	78.00	
Garden size, sq. m (Chi-S *p*-value ^2^)	**(0.033)**	**(0.025)**	(NS)	**0.029**
under 50	4.84 ± 6.63	0.00	9.38	
50–100	12.90 ± 13.11	3.33	21.88	
100–500	27.60 ± 8.10	33.33	21.88	
over 500	40.83 ± 22.39	56.67	25.00	
FGA (Chi-S *p*-value ^2^)	**(0.036)**	(NS)	(NS)	**0.038**
under 1/3	0.00	0.00	0.00	
1/3–1/2	29.17 ± 5.89	33.33	25.00	
over 2/3	56.56 ± 4.86	60.00	53.13	
Greenhouse (Chi-S *p*-value ^2^)	(NS)	(NS)	(NS)	NS
No	38.33 ± 11.79	46.67	30.00	
Yes	46.88 ± 4.42	43.75	50.00	
Age (Chi-S *p*-value ^2^)	(NS)	(NS)	(NS)	NS
35–55	3.50 ± 2.12	5.00	2.00	
56–75	12.00 ± 1.41	13.00	11.00	
76–84	7.00 ± 0.00	7.00	7.00	
Sex (Chi-S *p*-value ^2^)	(NS)	(NS)	(NS)	NS
Single female	38.00 ± 8.49	44.00	32.00	
Single male	44.44 ± 31.43	22.22	66.67	
Family	46.30 ± 13.09	55.56	37.04	
Persons involved in gardening per home garden (Chi-S *p*-value ^2^)	(NS)	(NS)	(NS)	NS
1	35.29 ± 0.00	35.29	35.29	
2	45.16 ± 4.56	48.39	41.94	
3 or more	46.43 ± 5.05	50.00	42.86	

^1^ Home gardens are grouped according to the settlement elevation: LL, lowland settlement ≤300 m a.s.l.; SMM, semi-mountainous or mountainous settlement over 300 m a.s.l. ^2^ Chi-Squared tests (Chi-S) were applied to access differences between individual home gardens. ^3^ Fisher’s exact tests were used to compare LL and SMM groups. NS, not significant (*p*-value > 0.05). SD, standard deviation. sq. m: square meters.

## Data Availability

The data presented in this study are available on request from the corresponding authors. The data are not publicly available so to protect the privacy of the studied participants.

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
