# Peer review of "Enough to Feed Ourselves!—Food Plants in Bulgarian Rural Home Gardens"

_plants, 2021, doi:10.3390/plants10112520_

Round 1

Reviewer 1 Report

This manuscript represents an ethnobotanical work on the Bulgarian rural home gardens. The study is well planned and the methods seem adequate and properly applied. The authors have made a census of the entities present in a significant sample of these gardens and by the way have carried out an analysis concerning their evolution and purposes, also highlighting their important role concerning both the transmission of knowledge, and the conservation of local food species.

Overall, I believe this is an interesting manuscript, that is potentially publishable. Before definite acceptance I propose a minor revision, for which I provide now the Authors with some comments.

  • The Authors of the entities should be written only the first time the species is mentioned; therefore the text (not tables) should be checked in order to respect this rule.

Furthermore, for some entities mentioned only once in the text (e.g. Castanea sativa, line number 293), the authors of the species is not really mentioned; obviously, they should be added.

  • For some species detailed with the subspecies or varieties (e.g. Raphanus raphanistrum sativus (L.) Domin, line number 214, Lactuca sativa varieties), the authors of the species should also be written both in the text and in Table 3.

  • Some inaccuracies should be corrected:

Cucurbitaceae (not Cucurbitacea, line number 20);

Capsella bursa-pastoris (L.) Medik. (not Medik, line number 171);

Allium tuberosum Rottler ex Spreng. (not Spreng, line number 211);

Malus domestica Borkh. (not Borkh, line number 281);

Eriobotrya japonica (Thunb.) Lindl. (not Lindl, line number 291).

Moreover, it’s better “Pomoideae and Prunoideae” (not Pomoidea and Prunoidea taxa, line number 283), as well as in Table 4.

  • In Table 4 the number 1 that appears in the caption must be added.

  • Information should be added on some characteristics of the study area (rainfall, temperature values, prevalent types of vegetation), which in similar works are usually debated in Materials and Methods ("Study area").

  • In the paragraph “Field Study and Research Approach”, the recent bibliographic reference used for the attribution of entities to the families should be indicated. The bibliographic sources used for the identification of the specimens also should be cited. Moreover, in the same paragraph (line number 546) number 3 should be incorrect.

  • As in the text (see previous comment), also in the supplementary table the authors of some species detailed with the subspecies or the variety are missing (e.g. Lactuca sativa crispa L., Origanum vulgare subsp. hirtum (Link) Ietsw., etc.); they should be added.

Moreover, some scientific names should be corrected:

Trigonella caerulea (not Trigonella caerullea);

Zea mays (not Zea mais).

Other annotation:

for Prunus cerasus, L. is the correct author, not Ledeb.;

it’s correct to write Prunus cerasus x P. avium, with the authors of the two species, and Prunus persica (L.) Batsch var. nectarina (Aiton) Maxim;

for Rosa sp. L. Cultivars, L. should not be indicated.

  • In the supplementary table, it’s reported that an extract (used for medicinal purposes) is obtained from six species, while in the paragraph “Plant Diversity and Crop Preference” it is written that the species used to obtain an extract are nine; this discrepancy should be corrected.

Author Response

Dear Reviewer 1,

Dear Editors,

Thank you very much for your encouragement to our attempt to explore the provisional role of home gardens from an ethnobotanical point of view using Bulgarian rural gardens as an example. Your positive outlook to our work gives us the confidence to continue studying the role of home gardening in food security of modern industrialized societies. We are grateful for your critical notes and suggestions how to improve the manuscript. We took them in consideration during the preparation of the revised version of the manuscript. You can see all changes and additions in the provided file in track changes.

Please find below our answers (in bold) to reviewer’s specific comments to the text:

1. Answer: We are very much obliged to you for pointing our mistakes and technical errors in the manuscript. The corrections you suggested can be seen in the provided file with track changes. Please see, the respective line numbers in the corrected version are added at the end of the each bullet number in bold script:

  • The Authors of the entities should be written only the first time the species is mentioned; therefore the text (not tables) should be checked in order to respect this rule. - The manuscript was rechecked and mistakes are corrected in the manuscript text and in the s1 table.
  • Furthermore, for some entities mentioned only once in the text (e.g. Castanea sativa, line number 293), the authors of the species is not really mentioned; obviously, they should be added. - Line: 316
  • For some species detailed with the subspecies or varieties (e.g. Raphanus raphanistrum sativus (L.) Domin, line number 214, Lactuca sativa varieties), the authors of the species should also be written both in the text and in Table 3. Line: 233 and Table 3 (page 6-7).
  • Some inaccuracies should be corrected:

         Cucurbitaceae (not Cucurbitacea, line number 20); Line: 23

         Capsella bursa-pastoris (L.) Medik. (not Medik, line number 171); Line: 187

         Allium tuberosum Rottler ex Spreng. (not Spreng, line number 211); Line: 230

         Malus domestica Borkh. (not Borkh, line number 281); Line: 304

         Eriobotrya japonica (Thunb.) Lindl. (not Lindl, line number 291). Line: 315

  • Moreover, it’s better “Pomoideae and Prunoideae” (not Pomoidea and Prunoidea taxa, line number 283), as well as in Table 4. Line: 315, table 4 (page 9-10)
  • In Table 4 the number 1 that appears in the caption must be added. Table 4 (page 9-10)
  • As in the text (see previous comment), also in the supplementary table the authors of some species detailed with the subspecies or the variety are missing (e.g. Lactuca sativa crispa L., Origanum vulgare subsp. hirtum (Link) Ietsw., etc.); they should be added. Line: 374-375
  • Moreover, some scientific names should be corrected:

         Trigonella caerulea (not Trigonella caerullea);

        Zea mays (not Zea mais).

        for Prunus cerasus, L. is the correct author, not Ledeb.;

        it’s correct to write Prunus cerasus x P. avium, with the authors of the two species,

         and Prunus persica (L.) Batsch var. nectarina (Aiton) Maxim;

         for Rosa sp. L. Cultivars, L. should not be indicated.

Thank you for these remarks. These will be provided in the final version of the Supplementary materials.

  • In the supplementary table, it’s reported that an extract (used for medicinal purposes) is obtained from six species, while in the paragraph “Plant Diversity and Crop Preference” it is written that the species used to obtain an extract are nine; this discrepancy should be corrected.

The correct number is six. Line: 207

  • Moreover, in the same paragraph (line number 546) number 3 should be incorrect. Line: 592

2. Answer: Thank you for suggesting additions to the Materials and Methods in order to improve replication and comparative research in the future. The respective line numbers in the corrected version are added here in bold script:

    • Information should be added on some characteristics of the study area (rainfall, temperature values, prevalent types of vegetation), which in similar works are usually debated in Materials and Methods ("Study area"). Lines: 542-555
    • In the paragraph “Field Study and Research Approach”, the recent bibliographic reference used for the attribution of entities to the families should be indicated. The bibliographic sources used for the identification of the specimens also should be cited. Lines: 597-598

Sincerely,

Teodora Ivanova and co-authors

12.11.2021

Reviewer 2 Report

This is an interesting paper presenting original data on ethnobotany from Bulgaria.

In my opinion, the paper needs some reorganization and improvement to become more incisive and clearer:

  1. a) The Introduction does not report a clear ‘Aim of the paper’, in the form of a couple of sentences that clearly explain why and where the research has been carried out. Usually, this part is at the end of the Introduction.
  2. b) I strongly recommend to move the Materials and Methods to the second chapter, just after the Introduction: this could clarify the meaning of some labels, the data sample studied and possibly the order of result presentation and discussion.
  3. c) The list of plants recorded must be entirely reported (currently only a small number is listed in Table 3), with the ‘use category’; the latter is the base for the histogram sums in Figure 1.

A different type of table could be prepared in the text to include more taxa, and a clear reference to the Supplementary Table 1 must be described in the paper (Methods section).

As a editing style, I suggest to left justify the first columns of the tables.

Line 20 =  Cucurbitaceae

Line 29 = crop and ornamental species

Line 31 = “the closest space of plant-human interaction – the home garden –“ = please, consider that this may be disputable, and that this is a rather recent historical condition

Line 36 = “Private gardens in the tropical zones” = it’s not so clear to me why the reference to the tropical zones is reported as prevalent; actually, the archaeobotanical research has often studied the plant remains from historical gardens and home waste deposits. As an example, I can remember the research made in the city of Ferrara, which is a UNESCO site for the Medieval/Renaissance home gardens preserved in northern Italy (Bosi et al. 2009 Luxury food and ornamental plants at the 15th century A.D. Renaissance court of the Este family (Ferrara, northern Italy. Vegetation history and archaeobotany18:389–402).

You can also compare, in the discussion, some of you results with the archaeobotanical evidence from Bulgaria, such as Hrisrova1 et al. 2015 (Plant economy and vegetation of the Iron Age in Bulgaria: archaeobotanical evidence from pit deposits)

Line 50 = for the same reason, I suggest to delete “in the Global North” or make a more precise reference to Europe (Check also line 501 for ‘Global South’)

Line 57 = “have presented dissimilar images” = this sentence is vague, please exain what means ‘dissimilar images’

Line 92 = SMM ?

Line 159 = in the paragraph 2.2 on Plant diversity, it is not clear the order of presentation of data

Line 199 = the latin has not plural with ‘s’ (alliums does not exist), but ‘Allia’; in any case you must refer only  to the genus name ‘Allium’

Delete the s from ‘Alliums’ also in the Fig. 1

Line 202 = (A. cepa L. Cepa Group) = what means ‘Cepa group’ ? is this a variety? Can you cite the botany nomenclature reference flora? Delete the same words at line 203.

Fig. 1 = can be reduced, reported in the same page (this is useful to compare histograms) and prepared in colours

Table 4 = Pomnoideae and Prunoideae (the latin names need ‘ae’)

Line 406 = Bulgaria; home = Bulgaria, home

Author Response

Dear Reviewer 2,

Dear Editors,

Thank you very much for your valuable suggestions to our efforts to explore the provisional role of home gardens from an ethnobotanical point of view using Bulgarian rural gardens as an example. We have considered your suggestions how to improve the manuscript adding the required information and correcting some flaws. Please find below our answers (in bold script) to reviewer’s specific comments to the text and find all corrections and additions in the uploaded file in track changes:

a) The Introduction does not report a clear ‘Aim of the paper’, in the form of a couple of sentences that clearly explain why and where the research has been carried out. Usually, this part is at the end of the Introduction.

Answer: Thank you for this remark; we added suggested text in the Introduction – see lines 94-97 

b) I strongly recommend to move the Materials and Methods to the second chapter, just after the Introduction: this could clarify the meaning of some labels, the data sample studied and possibly the order of result presentation and discussion. As a editing style, I suggest to left justify the first columns of the tables.

Answer: We used the template provided by the Journal and followed the guidelines for the preparation of the manuscripts where the Materials and Methods section is placed after the Discussion.

c) The list of plants recorded must be entirely reported (currently only a small number is listed in Table 3), with the ‘use category’; the latter is the base for the histogram sums in Figure 1. A different type of table could be prepared in the text to include more taxa, and a clear reference to the Supplementary Table 1 must be described in the paper (Methods section).

Answer: Thank you for these suggestions. We prepared Supplementary Table 1 in that way because the documented taxa are 145 and the full list spreads on 10 pages. The purpose of Table 3 was to present the taxa found in most of the gardens in all provinces. In response with your suggestion, we added two new columns: “use category” (Consumption) and also Preservation in the table 3 – see pages 6-7. Table 3 and figure 1 are not related. The latter presents the areas in the garden occupied by the certain crop groups. We propose to add the exact data for preparation of figure 1 as supplementary table 2, which can be found at the end of this file (page 22).

Line 20 =  Cucurbitaceae;

Answer: Thank you for the correction, see line: 23

Line 29 = crop and ornamental species;

Answer: Thank you for the correction, see line: 32

Line 31 = “the closest space of plant-human interaction – the home garden –“ = please, consider that this may be disputable, and that this is a rather recent historical condition .

Answer: It is changed to “one of the closest spaces of plant-human interaction – the home garden “ - line: 34

Line 36 = “Private gardens in the tropical zones” = it’s not so clear to me why the reference to the tropical zones is reported as prevalent; actually, the archaeobotanical research has often studied the plant remains from historical gardens and home waste deposits. As an example, I can remember the research made in the city of Ferrara, which is a UNESCO site for the Medieval/Renaissance home gardens preserved in northern Italy (Bosi et al. 2009 Luxury food and ornamental plants at the 15th century A.D. Renaissance court of the Este family (Ferrara, northern Italy. Vegetation history and archaeobotany18:389–402). You can also compare, in the discussion, some of you results with the archaeobotanical evidence from Bulgaria, such as Hrisrova1 et al. 2015 (Plant economy and vegetation of the Iron Age in Bulgaria: archaeobotanical evidence from pit deposits)

Answer: We are grateful for this suggestion. Our study was focused mostly on the present structure, plant diversity and function of home gardens. That is why in the introduction we included mostly papers concerning the food provisional role of the home garden in the recent years. We appreciate your suggestion to make comparisons with archaeobotanical data and made additions to the Introduction and Discussion sections. Please, see lines 42-44 and 511-519 in the provided revised version of the manuscript. Of course this important subject deserves additional study and analyses in further research.

Line 50 = for the same reason, I suggest to delete “in the Global North” or make a more precise reference to Europe (Check also line 501 for ‘Global South’)

Answer: Please, see provided rephrasing on lines 41, 56, 533.

Line 57 = “have presented dissimilar images” = this sentence is vague, please exain what means ‘dissimilar images’

Answer: The sentence was reformulated. Please, see line 63.

Line 92 = SMM ?

Answer: We use LL (lowland) and SSM (semi-mountainous and mountainous) codes for the settlements’ elevation LL < 300 m a. s.l. > SMM so to omit frequent repetitions throughout the paragraphs. We put additional clarification on lines 102-103

Line 159 = in the paragraph 2.2 on Plant diversity, it is not clear the order of presentation of data

Answer: We named subsection 2.2 “Plant Diversity and Crop Preference” because on one side the plant diversity in home gardens relies on the preference of the owners and on the other – most of the plant taxa we recorded were crops.

Line 199 = the latin has not plural with ‘s’ (alliums does not exist), but ‘Allia’; in any case you must refer only  to the genus name ‘Allium’. Delete the s from ‘Alliums’ also in the Fig. 1

Answer: Thank you for your remark. Please, see provided corrected version, line: 217

Line 202 = (A. cepa L. Cepa Group) = what means ‘Cepa group’ ? is this a variety? Can you cite the botany nomenclature reference flora? Delete the same words at line 203.

Answer: We appreciate this comment. We included the horticultural groups, based on morphological characteristics and growth habit of the plants, which are commonly used in horticultural/agricultural studies of the A. cepa varieties so to make the paper more useful for a wider audience. This distinction is informal, still it provides additional information on the diversity and characteristics of local landraces, likely unknown to the reader. Respective additional information is put in the Material and Methods section, see lines 598-600. The following references provide material on the subject:

Jones, H.A. and L.K. Mann. 1963. Onions and their allies: botany, cultivation, and utilization. Leonard Hill Books Limited, London.

https://www.bogos.uni-osnabrueck.de/index.php?cat=Publikationen&file=Ch1Taxonomy_and_Evolution_Allium.pdf

Hanelt, P., Schultze-Motel, J., Fritsch, R., Kruse, J., Maaß, H.I., Ohle, H. and Pistrick, K. (1992) Infrageneric grouping of Allium – the Gatersleben approach. In: Hanelt, P., Hammer, K. and Knüpffer, H. (eds) The Genus Allium – Taxonomic Problems and Genetic Resources. Proceedings of an International Symposium, Gatersleben, 11–13 June 1991. IPK, Gatersleben, Germany, pp. 107–123.

Fig. 1 = can be reduced, reported in the same page (this is useful to compare histograms) and prepared in colours

Answer: We reduced the size of the figure to fit in the page and add color to it.

Table 4 = Pomnoideae and Prunoideae (the latin names need ‘ae’)

Answer: Please, see provided corrected version of the manuscript – Table 4 (pages 9-10)

Line 406 = Bulgaria; home = Bulgaria, home

Answer: Please, see provided corrected version of the manuscript. Line: 429

Sincerely,

Teodora Ivanova and co-authors

12.11.2021

Reviewer 3 Report

Dear Authors,
I have greatly appreciated your work which appears complete in every section. The introduction well illustrates the evolution of rural area agriculture in relation to more industrialized agriculture in more recent times. The methodology is appropriate, the submission of the questionnaires is balanced, and the statistical analysis follows up-to-date and functional protocols.

I think, therefore, your work is very well done and I think it is the first step for a more consistent action for a recovering of old varieties of vegetables, the dissemination of the harvest and conservation of seeds, and the valorization of traditional varieties in the development of social policies related to the preservation of traditional food preparations that provide tools for food security, especially in rural areas of any country.

Just a few comments:
(a) in the abstract, some sentences should be added in relation to agricultural biodiversity because in that section it seems that it is not completely taken into account. Talking about traditional food is not enough because in many rural areas traditional food is prepared with non-traditional varieties, losing important historical links and a lot of agrobiodiversity.
(b) if available you should add some data on the level of education of the respondents because some technical choices in the management of family gardens could be due to adequate or poor training on the subject.
(c) you should also add some more details on the presence or absence of technical assistance in the areas where the questionnaires were administered.  There is often talk of crop rotations, sowing of seeds from a technical point of view, etc.
(d) if available, you should add some more details in relation to the prevalent seasonality of family gardens: winter? summer?

Regards

Author Response

Dear Reviewer 3,

Dear Editors,

Thank you very much for your encouraging words to our attempt to explore the provisional role of home gardens from an ethnobotanical point of view using Bulgarian rural gardens as an example. Your positive outlook to our work makes us hopeful that home garden research could contribute to development of more effective policies and common relevant actions towards preservation of (agro)biodiversity even in industrialized countries. We have considered your suggestions how to improve the manuscript and you can see all changes and additions in the provided file in track changes.

Please, find below our answers to reviewer’s specific comments to the text in bold script:

(a) in the abstract, some sentences should be added in relation to agricultural biodiversity because in that section it seems that it is not completely taken into account. Talking about traditional food is not enough because in many rural areas traditional food is prepared with non-traditional varieties, losing important historical links and a lot of agrobiodiversity.

Answer: We included one more sentence on this account as the abstract is limited to “about 200 words”. See lines 11-13.

(b) if available you should add some data on the level of education of the respondents because some technical choices in the management of family gardens could be due to adequate or poor training on the subject.

(c) you should also add some more details on the presence or absence of technical assistance in the areas where the questionnaires were administered.  There is often talk of crop rotations, sowing of seeds from a technical point of view, etc.

Answer (b+c): Thank you for the suggestions. As we mentioned in the Discussion our participants “utilized broadly the advancements in agronomy picked up from the state agricultural complexes that managed agriculture in Bulgaria during Communism.” This is related not only to the opportunities for seasonal and/or permanent occupation in agriculture typical in Bulgarian rural areas but also to obligatory use of unpaid labor of students and workers during that era. Hence, the obtained skills likely contributed to the developments and changes in the gardens. However, the impact of work-based learning and formal/informal education in agriculture should be the objective of additional studies, especially in relation to the transformation of traditional knowledge. Only two of our respondents had specialized education in agri/horticulture and we added this information in the Results – see lines 173-176.

 (d) if available, you should add some more details in relation to the prevalent seasonality of family gardens: winter? summer?

Answer: Climate in Bulgaria is predominately Continental with four well distinguished seasons. The winter is typically the resting period for gardening activities. See added text in the Results – lines 155-158.

Sincerely,

Teodora Ivanova and co-authors

12.11.2021

Round 2

Reviewer 2 Report

I am happy with the corrections made by the Authors, and I hope that they are happy too about the improvements made.

I have only still concerns about the Figure 1, because it was no matter of reduction of the size of the previous diagrams: the figure must be combined in a new version and the label must be rewritten to maintain clarity.